# A Simple Photonic System for DFS and AOA Simultaneous Measurement

**Xintong Li** [1,2,†] [ID]**, Jinming Tao** [1,2,†] [ID]**, Jinye Li** [1,†]**, Qianqian Jia** [1,2] [ID]**, Chaoquan Wang** [1,3] **and Jianguo Liu** [1,2,*]

[1] The State Key Laboratory of Integrated Optoelectronics, Institute of Semiconductors, Chinese Academy of Sciences, Beijing 100083, China

[2] The College of Materials Science and Opto-Electronic Technology, University of Chinese Academy of Sciences, Beijing 100049, China

[3] School of Integrated Circuits, University of Chinese Academy of Sciences, Beijing 100049, China

[*] Correspondence: jgliu@semi.ac.cn

[†] These authors contributed equally to this work.

**Abstract:** A simple photonics-based dual-channel system is proposed to simultaneously measure the Doppler frequency shift (DFS) and angle of arrival (AOA) of microwave signals. The system applies two parallel push–pull Mach–Zehnder modulators (MZMs) for carrier suppression dual-sideband (CS-DSB) modulation. The introduction of the reference signal results in a DFS measurement without direction ambiguity. The DFS can be determined by measuring the frequency of the down-converted intermediate frequency (IF) signal, and the AOA can be calculated by comparing the phase shift of the two channels. A proof-of-concept experiment shows that the DFS measurement error is less than 0.4 Hz during $\pm 100$ kHz, and the AOA measurement error is within $1.5°$ in a range of $0$–$70°$.

**Keywords:** Doppler frequency shift; angle of arrival; microwave photonics

## 1. Introduction

Obtaining the position and speed of moving objects is of vital importance in radar ranging, intelligent driving, wireless communication, and other emerging fields [1,2]. The measurement of these parameters can be translated into the measurement of the Doppler frequency shift (DFS) and angle of arrival (AOA), which are usually based on electrical systems. However, due to the limitations of electromagnetic interference, operating bandwidth, and other electronic bottlenecks, electrical methods are difficult to implement when the frequency to be measured is in the range of MHz to GHz. Therefore, more and more related research focuses on the method of microwave photonics to take advantage of both electrical and optical advantages [3–5].

At present, there are many photonics-related methods for measuring the DFS [6–12] and AOA. The measurement of the DFS can usually be obtained from the down-converted intermediate frequency (IF) signal. The direction of the DFS can be obtained by comparing the frequency between the output signal and the reference signal [9–12]. The measurement of the AOA is mainly achieved by measuring the relative time delay [13–15] or phase difference [16–20] of the echo signals received by the antennas. In [15,20], the measurement of the phase difference is converted to the measurement of electrical signal power. However, this scheme has very high requirements for the carrier suppression ratio and modulator bias point.

The above scheme can only measure one of either the DFS or AOA. Recently, schemes to measure these two parameters simultaneously have also been reported. In [21,22], researchers use a dual-channel microwave photon mixer and two PDs to obtain two IF signals, and then they obtain the DFS and AOA at the same time by measuring the frequency and phase difference of the electrical signals. However, these schemes require filters to remove excess sidebands. In [23], the optical signal is divided into two channels with a

wavelength-division multiplexer (WDM) at the output end of the dual-parallel MZM, and the DFS and AOA can be obtained by comparing the two channels of beat frequency signals. Ref. [24] implements a simple link system by applying two dual-parallel MZMs, in which the DFS value can be directly detected while the DFS direction and AOA can be obtained by comparing the phase of the two down-converted signals. Moreover, the system needs an additional bias point control circuit to ensure the modulator is implemented in carrier suppression single-sideband status. In [25–27], the DFS is calculated from the frequency of the IF signal by adding a reference signal, and the AOA is obtained by measuring the power of the IF signal.

In this paper, a simple photonics method for the simultaneous measurement of the DFS and AOA is presented. In this system, the optical carrier is divided into two channels and sent to two MZMs. A pair of reference signals are fed into the radio frequency (RF) ports of two push–pull MZMs after combining them with two echo signals. Both MZMs realize the carrier suppression dual-sideband (CS-DSB) modulation. The value of the DFS without directional ambiguity can be calculated from the frequency of the IF signal, and the AOA can be measured by comparing the phase difference between the two branches. Because no filter or WDM is involved, the scheme has a simple and compact configuration with a large working bandwidth. Moreover, the system does not need to control the modulator bias point precisely because electrical signal power mapping is not involved, which is convenient for practical application. As a verification of the system, when the echo signal frequency is around 15 GHz, the DFS measurement error of the system is within $\pm 0.4$ Hz, and the AOA measurement error is less than $1.5°$ from $0°$ to $70°$.

## 2. Principle and Method

The schematic diagram of the simulated DFS and AOA measurement proposed in this paper is depicted in Figure 1. The continuous wave (CW) emitted by the laser diode is divided into two channels and injected into two push–pull MZMs after adjusting the polarization state by two polarization controllers (PCs). Two echo signals of equal frequency are mixed with the reference signal and then fed into the two MZMs, respectively. The phase delay between the two echo signals is caused by the antenna spacing and AOA, which can be expressed as:

$$\tau = \frac{d \sin \alpha}{c} \tag{1}$$

$$\theta = \omega_e \tau \tag{2}$$

where $d$ is the distance between the two antennas, which is generally designated as $\lambda/2$, $c$ is the speed of light in a vacuum, $\tau$ is the time delay, $\alpha$ is the AOA of the echo signal, $\omega_e$ is the angular frequency of the echo signal, and $\theta$ is the phase difference between the two echo signals. Thus, the AOA can be calculated by:

$$\alpha = \arcsin\left(\frac{c\tau}{d}\right) = \arcsin\left(\frac{c(\theta/\omega_e)}{\lambda/2}\right) = \arcsin\left(\frac{\theta}{\pi}\right) \tag{3}$$

In the proposed system, the reference signal generated by the microwave source and the two echo signals received by the antenna can be expressed as:

$$\begin{bmatrix} \text{RF} \\ \text{Echo1} \\ \text{Echo2} \end{bmatrix} = \begin{bmatrix} V_r \sin(\omega_r t) \\ V_e \sin(\omega_e t) \\ V_e \sin(\omega_e t + \theta) \end{bmatrix} \tag{4}$$

where $V_r$ and $\omega_r$ are the amplitude and angular frequency of the reference signal, respectively, and $V_e$ and $\omega_e$ are the amplitude and angular frequency of the echo signal, respectively.

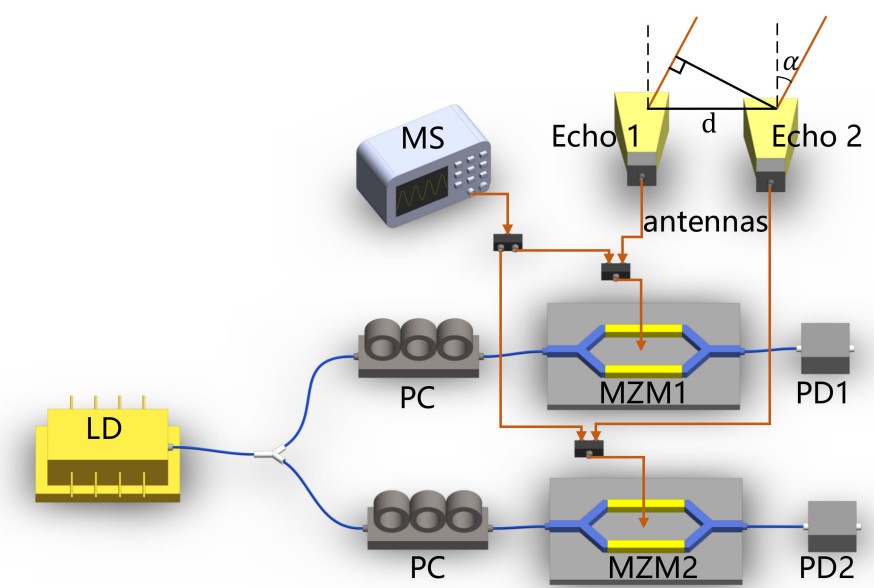

**Figure 1.** Schematic diagram of the proposed system for DFS and AOA measurement. LD: laser diode; PC: polarization controller; MZM: Mach–Zehnder modulator; PD: photodetector; MS: microwave source.

According to [9,12], the DFS is usually within ±1 MHz in most applications. In addition, as shown in Figure 2a, a direct down-converted microwave signal ($f_t - f_e$) will lead to measurement ambiguity. Therefore, we set the reference signal frequency to the transmitted signal plus an additional frequency $f$. When $f$ is greater than 1 MHz, the ambiguity will be eliminated, so that the down-converted signal ($f_r - f_e$) can be used to define whether the DFS is positive or negative as displayed in Figure 2b. Here, $f$ is set to 3 MHz, and the relationship among these frequencies can be shown as:

$$f_{DFS} = f_e - f_t \tag{5}$$

$$f_r = f_t + 3\,\text{MHz} \tag{6}$$

$$f_{IF} = f_r - f_e \tag{7}$$

where $f_{DFS}$, $f_e$, $f_t$, and $f_r$ are the DFS, echo signal frequency, transmit signal frequency, and reference signal frequency, respectively. When $f_{IF}$ is determined, $f_{DFS}$ has a unique corresponding value, which can be given by

$$f_{DFS} = 3\,\text{MHz} - f_{IF} \tag{8}$$

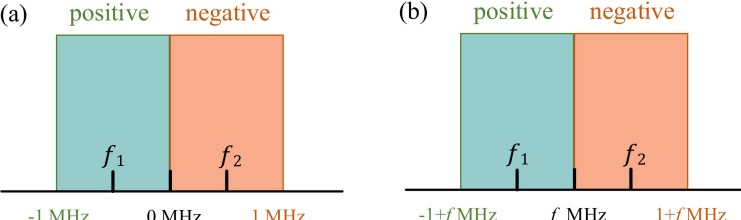

**Figure 2.** Down-converted microwave signal (**a**) with directional ambiguity; (**b**) without directional ambiguity by adding a reference.

Besides value, the direction of the DFS can be determined by measuring the frequency of the IF. When $f_{IF}$ ranges from 2 to 3 MHZ, the DFS is positive, and the target is moving

toward the radar. When $f_{IF}$ is 3–4 MHZ, the DFS is negative, and the target is moving away from the radar.

The reference signal and one of the echo signals are merged to drive MZM1. MZM1 is modulated in carrier-suppressed double sideband (CS-DSB), and the electrical field of output light can be expressed as:

$$E_1(t) = \frac{1}{4}E_0 e^{j\omega_c t} \left[ \begin{array}{c} e^{j\beta_e \sin(\omega_e t) + j\beta_r \sin(\omega_r t)} \\ +e^{j\beta_e \sin(\omega_e t + \pi) + j\beta_r \sin(\omega_r t + \pi)} e^{j\varphi_1} \end{array} \right] \tag{9}$$

where $E_0$ and $\omega_c$ are the magnification and angular frequency of the optical carrier, $\beta_e = \pi V_e / V_\pi$ and $\beta_r = \pi V_r / V_\pi$ are the modulation index of MZM1, and $\varphi_1$ is the optical phase shift caused by the direct current (DC) bias of the MZM1, which equals to $\pi V_{DC1} / V_\pi$. Setting MZM1 at the null point ($\varphi_1 = \pi$), applying the Jacobi–Anger expansion, and ignoring higher-order sidebands ($\geq 2$), Equation (9) can be rewritten as:

$$E_1(t) = \frac{1}{2}E_0 e^{j\omega_c t} \left[ \begin{array}{c} J_0(\beta_r)J_1(\beta_e)\left(e^{j\omega_e t} + e^{-j\omega_e t}\right) \\ +J_0(\beta_e)J_1(\beta_r)\left(e^{j\omega_r t} + e^{-j\omega_r t}\right) \end{array} \right] \tag{10}$$

where $J_n(\beta)$ is the nth-order Bessel function of the first kind.

There is a phase difference $\theta$ between the two echo signals. The reference signal is mixed with the other echo signal and injected into MZM2. The DC bias of MZM2 is also set at the minimum transmission point. Using the Jacobi–Anger expansion and ignoring higher-order terms, the electric field at the output of the MZM2 can be written as:

$$E_2(t) = \frac{1}{2}E_0 e^{j\omega_c t} \left[ \begin{array}{c} J_0(\beta_r)J_1(\beta_e)\left(e^{j(\omega_e t + \theta)} + e^{-j(\omega_e t + \theta)}\right) \\ +J_0(\beta_e)J_1(\beta_r)\left(e^{j\omega_r t} + e^{-j\omega_r t}\right) \end{array} \right] \tag{11}$$

After that, the two optical signals are respectively fed into two low cutoff-frequency PDs for beat processing to eliminate high-order sidebands. Then, the two IF signals are sent to an electrical spectrum analyzer (ESA) for analysis. The DC terms can be directly ignored because the related frequency range is 2–4 MHz. The photocurrent from the two PDs can be expressed as:

$$\begin{aligned} I_1(t) &\propto \eta E_0^2 J_0(\beta_e)J_0(\beta_r)J_1(\beta_e)J_1(\beta_r)\cos[(\omega_r - \omega_e)t] \\ &= \eta E_0^2 J_0(\beta_e)J_0(\beta_r)J_1(\beta_e)J_1(\beta_r)\cos(\omega_{IF}t) \end{aligned} \tag{12}$$

$$\begin{aligned} I_2(t) &\propto \eta E_0^2 J_0(\beta_e)J_0(\beta_r)J_1(\beta_e)J_1(\beta_r)\cos[(\omega_r - \omega_e)t - \theta] \\ &= \eta E_0^2 J_0(\beta_e)J_0(\beta_r)J_1(\beta_e)J_1(\beta_r)\cos(\omega_{IF}t - \theta) \end{aligned} \tag{13}$$

where $\eta$ is the responsivity of the PDs. The frequency of the IF signal can be read directly from the ESA, and then the value and direction of the DFS can be determined. The AOA can be obtained by comparing the phase relationship between the two branches.

### 3. Experiment Result

A verification experiment is set up based on Figure 1. The laser diode (LD, Santec TSL-550) emits a linearly polarized light as the optical carrier. The optical carrier with a center wavelength of 1550 nm and a power of 13 dBm is divided into two channels, passed through two PCs, and then injected into two MZMs (EOSPACE AZ-DV5-65, $V_\pi \sim 7$ V). A 15 GHz $\pm$ 1 MHz microwave signal is generated from a microwave source (KEYSIGHT E8267D) with a power of 10 dBm, and then divided into two equal channels as two echo signals. The phase difference between the two echo signals is provided by an additional phase shifter (CONNPHY MPS, DC-18G) in one of the channels. The 15 GHz + 3 MHz microwave signal generated by another microwave source (Ceyear 1464A) is used as the reference signal, which is divided into two channels and then connected with two echo signals, respectively, by electrical couplers (Talent Microwave RS2W10400, 1–40 GHz). The

two combined microwave signals are used to drive two MZMs, respectively, and both push–pull MZMs are biased at the minimum transmission point. The two MZMs are followed by two low-frequency PDs (THORLABS, DET01CFC/M) with a cut-off frequency of 2 GHz, and then an oscilloscope with a sampling rate of 2.5 GSa/s (ROHDE&SCHWARZ RTB2002) is used to obtain the IF signal frequency and the phase difference between the two branches. In addition, an ESA (ROHDE&SCHWARZ FSV) is used to measure the electrical spectrum, while an optical spectrum analyzer (OSA, Anritsu MS9740B) is used to observe the optical spectrum and monitor the MZMs' bias state.

The ESA is applied to observe the electrical spectrums. The resolution bandwidth (RBW), video bandwidth (VBW), and scan range are set to 1 kHz, 1 kHz, and 4 MHz, respectively. Figure 3 depicts the output electrical spectrum of the IF signal. When the frequency of the output IF signal is 2.5 MHz, the corresponding DFS is +0.5 MHz, which means that the target is approaching the receiver. If the echo signal is adjusted from 15 GHz + 0.5 MHz to 15 GHz − 0.5 MHz, the peak frequency of the output electrical spectrum is 3.5 MHz, which implies that a negative DFS equal to −0.5 MHz is obtained, and the simulation target is moving away from the receiver.

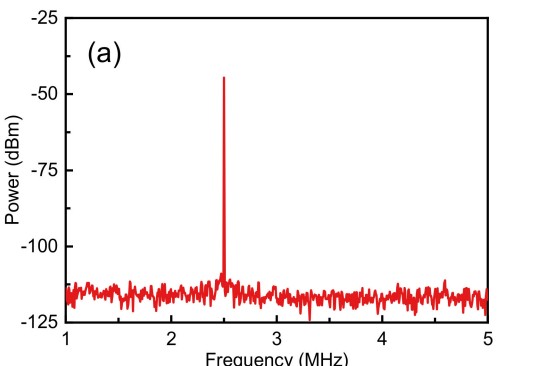 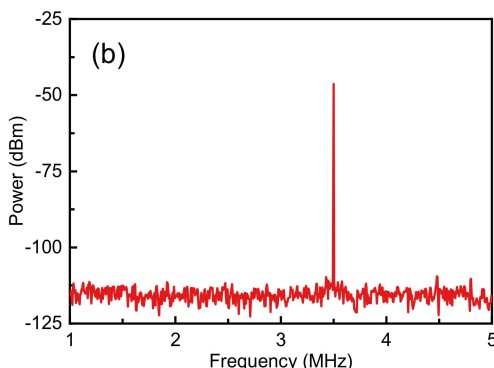

**Figure 3.** Measured system output electrical spectrum for (**a**) a +0.5 MHz DFS and (**b**) a −0.5 MHz DFS.

To further verify the DFS measurement system within a wide spectrum range, the frequency of the echo signals is changed from a range of 15 GHz − 100 kHz to 15 GHz + 100 kHz with a step of 10 kHz. The ESA RBW, VBW, and scan range are set to 1 Hz, 1 Hz, and 10 Hz, respectively. As shown in Figure 4, the DFS measurement result fits well with the theoretical value, with errors within ±0.4 Hz.

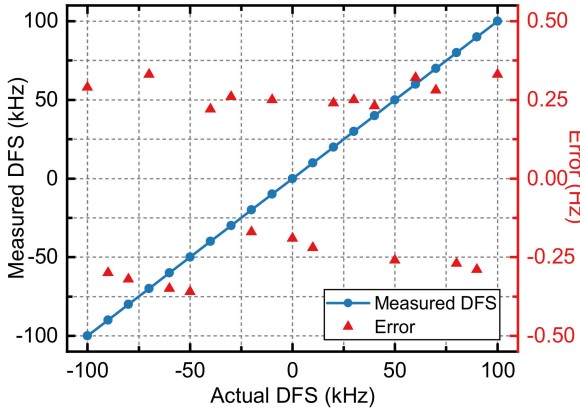

**Figure 4.** Measured DFS (blue circular line) and corresponding errors (red triangular) for the DFS changed from −100 kHz to 100 kHz with a step of 10 kHz.

Figure 5 shows the electrical waveforms of two branches with the same initial phase. In Figure 5a (or Figure 5b), the phase difference and the power difference between the two branches are caused by the incomplete consistency of the system, and the inherent

difference can be easily eliminated. When the phase difference introduced by the phase shifter is 0°, the phase difference between the two signals is recorded. At this time, the phase difference is the inherent phase difference caused by the inconsistency of the two test branches. Then, the phase of the lower branch is changed by the phase shifter, and the phase differences between the corresponding two branches are recorded. The actual measured phase differences are equal to the recorded phase differences minus the inherent phase difference. Compared to Figure 5a,b, the output power is different with different measurement frequencies, which is introduced by the bias point drift of the modulators. Our scheme does not involve power mapping, so the influence of power difference on the experimental results is negligible.

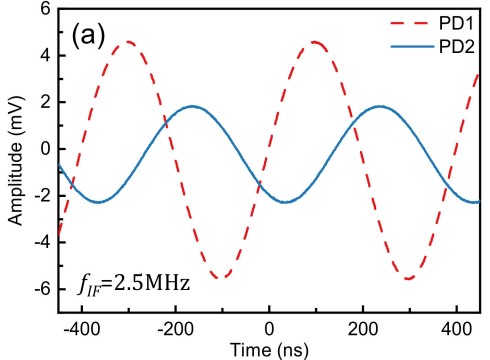 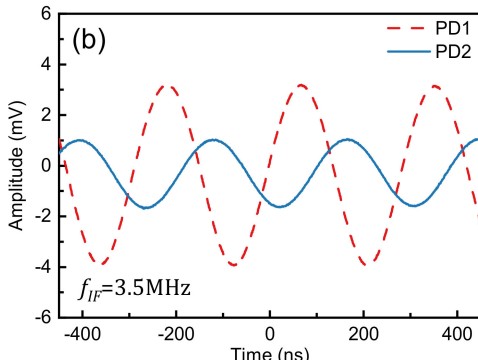

**Figure 5.** Electrical waveforms of down-converted IF signal when the DFS is (**a**) +0.5 MHz and (**b**) −0.5 MHz, respectively.

Figure 6 illustrates the relationship between the AOA and the phase difference between two echo signals. It can be seen that as the phase difference moves toward 180°, a slight change will cause a great shift in the AOA. Therefore, it is unnecessary and inaccurate to measure the AOA by the phase shift in this interval. In order to further demonstrate the AOA measurement capability of the system, the frequency of the echo signals is set to 15 GHz + 0.5 MHz, and the phase difference is changed to a range of 0~180° with a phase shifter. The phase shifter used in this experiment is a coaxially manually adjusted phase shifter, and the phase shifter value can be derived directly from the working frequency band and the reading of the phase shifter. However, given the inherent manufacturing error of the device, the value derived from the reading may not be accurate.

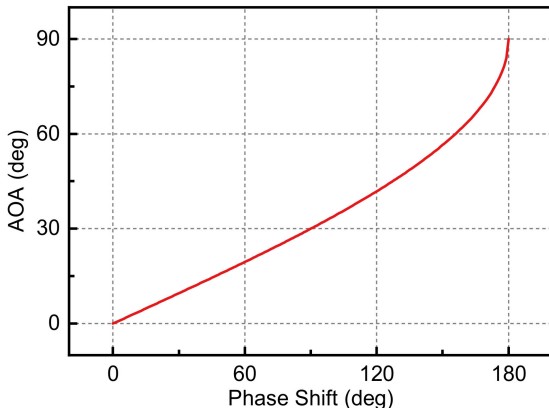

**Figure 6.** AOA versus the echo signal phase shift from 0° to 180°.

Therefore, as shown in Figure 7a, an electrical vector network analyzer (KEYSIGHT N5227B) is used to measure the phase shift (blue circular solid line), and the results are compared with the proposed scheme (red circular solid line). The measurement error of the two methods (green triangular solid line) is less than ±4.5° in the range of 0~170°.

According to Equation (3), the phase shift is converted into the AOA displayed in Figure 7b. Similarly, the blue circular solid line, the red circular solid line, and the green triangular solid line represent the estimated AOA value by EVNA, by the proposed scheme, and the measurement error of the two methods, respectively. It can be seen that the AOA measurement error is within ±1.5° from 0° to 66.44°.

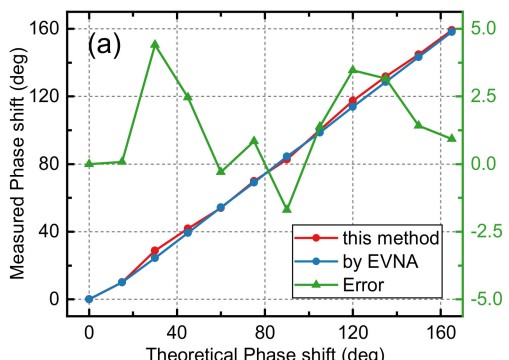 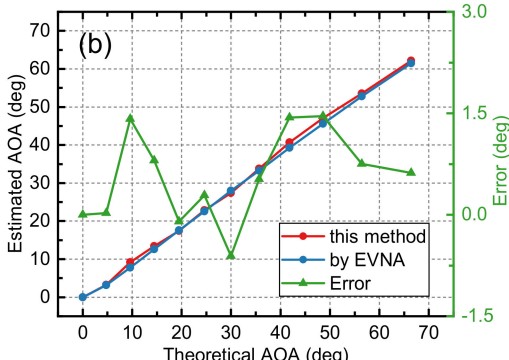

**Figure 7.** (**a**) Phase shift and (**b**) AOA measured by this method, EVNA, and the corresponding measurement error.

## 4. Discussion

A simultaneous measurement system for the DFS and AOA based on photonics is proposed in this paper. With the introduction of the reference signal, the direction and value of the DFS can be easily determined from the output signal. There is a one-to-one correspondence between the AOA and the phase difference between the two branches from 0° to 180°, and the theoretical measurement range of the AOA is from 0° to 90°. However, as shown in Figure 6, the mapping relationship between the AOA and the phase difference is not linear. When the phase difference is close to 180°, a slight phase difference change will result in a large AOA change. Therefore, for the same phase difference measurement error, when the phase difference is close to 180°, the corresponding AOA measurement error will be larger. The measurement is performed at a step size of 15° within a phase difference range of 0° to 180° in Figure 7a, and the corresponding results are converted to the measurement for the AOA as shown in Figure 7b. Due to the large measurement error for the AOA at 90°, it is abandoned. The actual phase difference measurement range is from 0° to 165°, and the corresponding AOA range is 0° to 66.44°. An increase in the actual measurement range of the AOA can be achieved by reducing the point-taking interval within the phase difference range of 165° to 180°.

## 5. Conclusions

As a result, a simple photonics-based DFS and AOA measurement system is proposed. The system uses two push–pull MZMs and constructs two optical branches. The DFS can be calculated by utilizing the frequency of the down-converted IF signal, and the AOA can be obtained by comparing the phase difference between the two branches. Experimental results display that the measurement error of the DFS is less than ±0.4 Hz for a 15 GHz echo signal with a frequency offset of ±100 kHz. After a simple calibration, the measurement error of the AOA is within ±1.5° in the range of 0~70°. In addition, our system has a large operating bandwidth because it does not involve optical filters or WDMs. More importantly, since no power mapping is involved, the change of the polarization state and the drift of the bias point have little effect on the experimental results. Our DFS and AOA measurement schemes have broad application prospects in electronic countermeasures, intelligent driving, man-machine confrontation, and so on.

**Author Contributions:** Conceptualization, X.L., J.T. and Q.J.; methodology, X.L. and J.T.; software, J.L. (Jinye Li) and Q.J.; validation, X.L., J.T. and C.W.; writing—original draft preparation, J.T. and J.L. (Jinye Li); writing—review and editing, X.L.; visualization, X.L. and J.L. (Jinye Li); project administration, J.L. (Jianguo Liu), X.L., J.T. and J.L. (Jinye Li) contributed equally to this work. All authors have read and agreed to the published version of the manuscript.

**Funding:** This work was funded by the National Natural Science Foundation of China under Grant 61727815.

**Acknowledgments:** The authors are grateful to the State Key Laboratory on Integrated Optoelectronics, Institute of Semiconductors, Chinese Academy of Sciences, and the College of Materials Science and Opto-Electronic Technology, University of Chinese Academy of Sciences.

**Conflicts of Interest:** The funders had no role in the design of the study; in the collection, analyses, or interpretation of data; in the writing of the manuscript; or in the decision to publish the results.

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
