# Peer review of "A Simple Photonic System for DFS and AOA Simultaneous Measurement"

_photonics, doi:10.3390/photonics9120980_

Round 1
Reviewer 1 Report
In the manuscript of photonics-2003315 submitted to the Photonics, the authors proposed a photonic approach for Doppler frequency shift (DFS) and angle of arrival (AOA) measurement. This work seems interesting and attractive, in which the DFS is calculated by the down-converted signals without direction ambiguity while the AOA is obtained by comparing the phase shift of two channels’ down-converted signals. Good performance in terms of DFS and AOA measurement errors with a wide measure range is verified by experiments.
All in all, the manuscript is well organized. I can recommend the manuscript for publication in PJ after the followings issues are addressed
1. “[…] is of vital importance in radar ranging, intelligent driving, wireless communication, and other emerging field” should be “[…] is of vital importance in radar ranging, intelligent driving, wireless communication, and other emerging fields”.
2. The full name of “LD” and “MSG” should be given in the text.
3. In order to highlight the article content, limitations of the measurement system would be better to be mentioned.
4. What do the blue circular and red triangular in Fig. 4 represent? Please give some detailed explanations.
Reviewer 2 Report
The authors propose a simple photonic system for DFS and AOA measurement. The principle and the results are clearly presented, however, the idea is relatively simple, it is based on the dual-channel microwave photonic mixing between the transmitted signal and the echo signals. Many works based on a similar idea were reported, and I cannot find something interesting or new from this manuscript, so I cannot recommend it to be accepted by MDPI Photonics.
Reviewer 3 Report
The authors demonstrated a filter-free photonic-based scheme to simultaneously measure the Doppler frequency shift (DFS) and the angle of arrival (AOA) of a target. In the scheme, the DFS is achieved by measurement of the intermediate frequency (IF) signal down-converted from the received signal. Meanwhile, according to the phase difference between the two IF signals from two down-conversion branches, the AOA can be calculated. In addition, the characteristic of large operation bandwidth is guaranteed by releasing the demand of wavelength selectors. The paper can be accepted for publication in Photonics if the authors can address the following comments properly.
1. Some setting parameters of the proof-of-concept experiment are missing. For example, the sampling rate of the oscilloscope is important for the measurement of the phase difference value. The power of two echo signals and the half-wave voltages of the MZM1 and MZM2 should be provided, for illustrating the modulation index of the electro-optics modulation. The resolution of the electrical spectrum analyzer presented in Fig. 3 should be supplemented.
2. The authors say that the phase and power imbalance between the two branches are calibrated. Thus, the calibration method should be illustrated in the paper.
3. The AOA in the range of 0° to 66.44° has been measured in the experiment. What is the maximal range of the proposed AOA measurement method? The reason why the maximal range has not been achieved in the experiment needs to be explained.
4. What’s more, some format errors and English writing problems should be addressed. Some of them are listed as following:
Page 6, line 201: “The phase shifter used in this experiment is coaxially manually adjusted, although the corresponding phase shift value can be read out, given its wide operating bandwidth characteristics and the inherent error of the instrument, the reading may not be accurate in the frequency band we have chosen.” This sentence has problems with grammar.
Page 4, line 194: “(a) phase shift” -> “(a) Phase shift”.
Page 3, line 99: “down-converted” -> “Down-converted”.
Reviewer 4 Report
The author proposed a method for Doppler frequency shift (DFS) and angle of arrival (AOA) measurement. The idea of this paper is very interesting. AOA is obtained by comparing the phase shift of downconverted signals of the two channels, which avoids the problem of inaccurate AOA measurement caused by offset point drift of the modulator. The experiment verifies the execellent performance of DFS and AOA measurement in a wide measurement range without offset point control.
The manuscript is well organized. After solving the following problems, I suggest that the manuscript be published in Photonics.
1. Line 117, “the DFS is negative, and the object is far away from the radar.” should to be “the DFS is negative and the target is moving away from the radar.”
2. Line 120, “the output light field” should be “the electrical field of output light”.
3. Line 171, “far away” should to be “moving away”.
4. Figure 4, please add legend.
5. I wonder if the measurement result is still accurate after adjusting the PC?
Reviewer 5 Report
The authors demonstrate a simple photonics-based dual-channel system is proposed to simultaneously measure the Doppler frequency shift (DFS) and angle of arrival (AOA) of microwave signals. The measurement accuracy of DFS and AOA is advanced compared with the current photonics-related methods. I am impressed by the high quality of the experimental measurement. I think the theme of this article is in line with the goal of the journal Photonics. Of course, there are some points that I suppose the authors should address to improve the manuscript further.
(1) In Figure 1, the author does not indicate which one is PD1 or PD2, so that the results of PD1 and PD2 in Figure 5 are somewhat ambiguous. I suggest the author add some explanations or mark them in figure 1.
(2) In line 180, the author declares that “the measurement error of DFS is within ±0.4 Hz, which fits well with the theoretical value”, but I’m confused that how the author obtained the theoretical value?
(3) Can the author explain how to effectively judge the direction of DFS by the method proposed in this paper? Because it is important to achieve the analysis of the moving direction of targets.
Round 2
Reviewer 2 Report
no further comments, it can be accepted now.